# Demystifying Inter-Class Disentanglement

**Aviv Gabbay**      **Yedid Hoshen**
School of Computer Science and Engineering
The Hebrew University of Jerusalem, Israel

## Abstract

Learning to disentangle the hidden factors of variations within a set of observations is a key task for artificial intelligence. We present a unified formulation for class and content disentanglement and use it to illustrate the limitations of current methods. We therefore introduce LORD, a novel method based on Latent Optimization for Representation Disentanglement. We find that latent optimization, along with an asymmetric noise regularization, is superior to amortized inference for achieving disentangled representations. In extensive experiments, our method is shown to achieve better disentanglement performance than both adversarial and non-adversarial methods that use the same level of supervision. We further introduce a clustering-based approach for extending our method for settings that exhibit in-class variation with promising results on the task of domain translation.

Project webpage: http://www.vision.huji.ac.il/lord

## 1 Introduction

Objects in the real world encompass many different attributes mixed together. Some of the attributes are permanent i.e. the class identity of the object, whereas others are transitory e.g. the pose of the object. Humans can often effectively separate between the class identity of the object, and the transitory pose of the object, even from a single observation. A key task for artificial intelligence is to empower computers to learn to separate between different attributes of observed data, often referred to as disentanglement. In this paper, we present a new method for achieving disentanglement between the class of an object and the sample-specific content. We restrict our attention to images, however some of our ideas may carry over to other modalities.

There are multiple settings for disentanglement. The simplest is fully supervised - for each training image both the class and content are given as labels. A fully supervised scheme (e.g. deep encoders) may be trained to recover the class and content information from a single image. Conversely, a generative model can be trained to generate an image given input class and content information. On the other extreme, fully unsupervised disentanglement takes as input a set of images with no further information. A successful unsupervised disentanglement algorithm will be able to learn a representation in which different factors of variation such as class and content will be represented separately. Fully unsupervised disentanglement is highly ambitious and is work in progress, current methods typically do not produce consistently good results in this setting (Locatello et al., 2019).

In this work, we deal with the class-supervised disentanglement task. In this setting, the class label for each image in the training set is given. Such supervision can be easily obtained in practice e.g. tracking an object in a video obtains multiple images in multiple poses of the same class (for example, person identity). The objective of the disentanglement task is to learn a representation containing all the information not available in the class label, denoted as content. In the case of faces, this content includes: head pose, facial expression, etc. We begin by carefully analyzing the information contained in the class and content representations. We show that current methods allow information to leak between the representations leading to imperfect disentanglement. We therefore introduce LORD, a novel method which carefully ensures no information leakage between the class and content representations.

Our method differs from previous methods by several methodological improvements. i) We leverage latent optimization to learn a single representation for each class which is shared between all

its samples. We show and discuss the benefits of this approach over the amortized techniques. ii) We introduce asymmetric regularization on the content latent codes to achieve class-invariant representations. We show the superiority of this technique over adversarial constraints and the KL-divergence. Latent optimization is very effective at learning disentangled representations at training time, however, it is not useful for obtaining class and content codes of unseen test images. Optimizing over the latent codes at test time (without class supervision which exists at training time), leads to overfitting which results in entangled representations. We overcome this challenge by introducing a second stage in which we use the class and content codes learned by our model in the first stage for training feed-forward class and content encoders. The encoders generalize well to unseen images and significantly reduce the inference time on new samples.

Our method is evaluated qualitatively and quantitatively in terms of generation of novel samples of observed classes. We also quantitatively evaluate the quality of disentanglement of learned features by classifying class labels from content codes and vice versa. Our method is shown to significantly outperform other adversarial and non-adversarial methods. Disentangling class and content representations assumes that intra-class variation is significantly lower than inter-class variation. We discover that this assumption can be relaxed by clustering in-class styles into separate classes. We demonstrate promising results of our approach on unsupervised domain mapping.

Our contributions in this work are as follows: i) An insightful analysis of class-conditional disentanglement. ii) LORD: a new well-motivated non-adversarial method for disentanglement achieving SOTA results by shared latent optimization and an asymmetric regularization. iii) Second stage amortization for single-shot class generalization. iv) The first effective method for disentanglement between 10k classes. v) A clustering based extension for style disentanglement.

## 1.1 RELATED WORK

Our work deals with class-supervised disentanglement. Several works based on variational autoencoders (VAEs) (Kingma & Welling, 2014) have attempted disentanglement with no supervision e.g. $\beta$-VAE (Higgins et al., 2017) and factor-VAE (Kim & Mnih, 2018). In an extensive comparative study, Locatello et al. (2019) show that none of the compared methods have been successful on all the datasets examined. It therefore seems likely that some supervision is required for effective disentanglement. Many works (including ours) provide only class supervision e.g. when the identity of a face is given but not its transitory attributes. The disentanglement between the factors of variation is enforced by adversarial constraints (Mathieu et al., 2016; Szabó et al., 2018; Denton & Birodkar, 2017; Hadad et al., 2018) or by non-adversarial constraints e.g. cycle (Harsh Jha et al., 2018) or variational group codes (Bouchacourt et al., 2018). Differently from the above works, we learn per-class codes rather than per-image class. Similar design choice were taken by cGAN and cVAE, but for the application of image generation rather than disentanglement. cGAN and cVAE require the latent space to be Gaussian, which hurts disentanglement performance.

Many disentanglement methods use adversarial training (Goodfellow et al., 2014). Success was achieved on image generation (Brock et al., 2019), image mapping (Isola et al., 2017) and domain alignment (Liu et al., 2017). Adversarial methods are notoriously hard to optimize, require very careful architecture and hyper-parameters tuning due to their min-max nature. To overcome these issues, non-adversarial methods have been proposed to achieve better results on tasks previously dominated by adversarial networks e.g. image synthesis (Bojanowski et al., 2018; Razavi et al., 2019), image-to-image mapping (Hoshen & Wolf, 2018) and word translation (Mukherjee et al., 2018). In this paper, we present a non-adversarial method achieving state-of-the-art performance on disentanglement.

## 2 CLASS AND CONTENT DISENTANGLEMENT

Assume that we are given a collection of $n$ images $x_1, x_2, ..., x_n \in \mathcal{X}$. For each image $x_i$, we are given a class label $y_i \in [k]$. We assume that every image belongs to a single class, although this requirement can be relaxed. Note that many images may share the same class label (e.g. faces of the same person at different poses). We denote the embedding of a given class $y$ as $e_y$. We assume that the images can be disentangled into representations in two latent spaces $\mathcal{Y}$ and $\mathcal{C}$. Therefore, our objective it to find a class representation $e_{y_i} \in \mathcal{Y}$ and a content representations $c_i \in \mathcal{C}$ for each

image $x_i$. Let us define the information that we wish each representation to contain. As there is some inconsistency in the notation used in the style-content, pose-content and domain translation literature, we will define our terms precisely.

The image class representation $e_{y_i}$, needs to include all information that is shared by all images sharing the same class e.g. if classes correspond to different facial identities, then the class representation must include all the time-invariant facial information. The content representation $c_i$ includes all the information that is unchanged if the image is transferred between classes. This information must be independent of the class-information. E.g for faces, content corresponds to time-varying facial information such as head pose and expression. Besides the class and content representations, images may contain other image-specific information, which is not represented by the class-label and is not expected to be transferred across classes. The difference between content and style is semantic and requires careful design. In the facial identity example the style may include noise, lighting conditions or nuisance background features. We denote the style representation as $s_i \in \mathcal{S}$.

We define a generator $G$, a neural network parameterized by $\theta$, which transforms the disentangled representations into an image. Given our definitions above each image can be modeled by:

$$x_i = G_\theta(e_{y_i}, s_i, c_i) \quad x_i \in \mathcal{X} \quad e_{y_i} \in \mathcal{Y} \quad s_i \in \mathcal{S} \quad c_i \in \mathcal{C} \tag{1}$$

The content must be independent of the class and style, however the style may be class dependent. E.g. if the classes are shoe images and edge images, styles within the shoes class may include particular colors and textures, which are typical of shoes but not of edge images. More formally, the mutual information between $c$ and $e_y$, $s$ should be zero:

$$I(c; e_y) = 0 \quad I(c; s) = 0 \tag{2}$$

In many cases, it can be assumed that inter-class variation is significantly larger that intra-class variation. Many approaches were devised to learn disentangled representations for this scenario, in which $s_i$ contains both class and style information of an image $x_i$. We will critically review several representative methods.

*Adversarial Methods*: One way to ensure the independence between the content and class/style representations is using adversarial discriminators. We will summarize the ideas proposed in DrNet (Denton & Birodkar, 2017) as this approach has the best performance of all adversarial methods. These techniques do not learn a class representation explicitly but instead strongly constrain a style encoding. The model of this method is described by:

$$x_i = G_\theta(0, s_i, c_i) \tag{3}$$

They attempt to ensure the similarity of styles of images in the class using a similarity constraint $\mathcal{L}_{similarity} = \|s_i - s_j\|^2$ if $y_i = y_j$. To ensure independence between $s$ and $c$, an adversarial discriminator $D_y(c_i, c_j)$ is trained to discover if two images are from the same class. If the content representation is truly disentangled, then no class information is available in the content code and the discriminator accuracy will not be greater than a random chance. This approach has two weaknesses: i) It does not directly prevent content information from leaking into the style representation (but only through a weak pairwise constraint). ii) Adversarial methods are notoriously hard to optimize and require careful hyper-parameter tuning due to the challenging saddle point optimization problem.

*Non-Adversarial Methods*: Due to the difficulty of adversarial training, non-adversarial methods have attracted attention. We will review the ideas in Multi-Level VAE (ML-VAE) (Bouchacourt et al., 2018), which performs the best of the non-adversarial methods and is most related to ours. ML-VAE also does not learn a class-representation, but a style representation $s_i$ via amortized inference. However, in order to limit the content information which flows to the generator from the style code, it relies on the presence of samples from the same class in a mini-batch during training and accumulates their style encodings using a product of normal densities before feeding the generator (the entire process is described in the original paper). To summarize, ML-VAE approximates the style representation $\bar{s}_M$ of a group of observations $M$ from the same class, and generates the image:

$$x_i = G_\theta\left(0, \bar{s}_{M_i}, c_i\right) \quad M_i = \{j | y_i = y_j\} \tag{4}$$

It limits the information in the content representation $c_i$ by constraining its distribution using KL-divergence with the standard normal distribution. This approach suffers from significant drawbacks: i) It uses grouped amortized encoding for inferring $\bar{s}_M$. As the size of a mini-batch is limited, this either limits the group accumulation to be over a few samples which is biased, or limits batch-diversity by only including a few classes which hurts optimization. ii) In our experiments, the KL-divergence does not sufficiently constrain the information in $c_i$ i.e. in practice class information is found in $c_i$.

## 3  LORD: LATENT OPTIMIZATION FOR REPRESENTATION DISENTANGLEMENT

In Sec. 2, we analyzed the task of disentanglement between class and content. Our analysis highlighted the issues faced by current state-of-the-art methods. In this section, we introduce a novel method motivated by the insights from the previous section.

### 3.1  LATENT OPTIMIZATION FOR CLASS SUPERVISION

We make explicit the assumption that inter-class variation is significantly larger than intra-class variation. This allows us to model images as a combination of class and content codes:

$$x_i = G_\theta(e_{y_i}, 0, c_i) \tag{5}$$

*Shared Latent Optimization*: We model the class representation as an embedding $e_y$ that is shared between all images belonging to the same class $\{x_i | y_i = y\}$. Instead of using amortized inference (learning a mapping from the image to the class codes using an encoder), we optimize over the class embeddings directly using latent optimization. This has several important benefits: i) As the code is shared exactly between all images belonging to the same class (each having different content), it is impossible to include any content information in the class code. ii) As we learn per-class representations directly rather than using previous techniques as group averaging, each mini-batch can contain images randomly sampled from all classes allowing maximal diversity.

We learn the content representation by optimizing over per-sample content embeddings directly using latent optimization and not in an amortized fashion using an image to content encoder. As we show in the experimental section, a model trained with latent optimization preserves a very high degree of disentanglement along the training and is less sensitive to hyperparameter choices.

*Asymmetric Noise Regularization*: Latent optimization over the class embeddings ensures that no content information is present in the class representation. To ensure that class information does not leak into the content representation, we regularize the content code to enforce minimality of information. Previous approaches attempted to minimize content information by setting a bottleneck of a small content code or by matching the content distribution to a prior normal distribution using KL-divergence. Using a small noiseless bottleneck, does not however reduce information significantly. A continuous variable may in fact store an infinite amount of information (although the amount of information the generator may extract is limited by other factors). In our experiments, we found that regularizing with KL-divergence (as done by previous works) led to a partial posterior collapse i.e. nearly all means and standard deviations learned by the encoder defaulted to $0$ and $1$ respectively, satisfying a perfect standard normal distribution. For a few components, the encoder learned large means and very small standard deviations. The KL-divergence therefore learned behavior similar to a small-size bottleneck. This phenomenon implies that regularizing the distribution of the content codes with KL-divergence may require additional attention and a careful hyperparameter tuning. We present the experimental evidence in the Appendix A.3.

In our approach, we regularize the content code with an additive Gaussian noise of a fixed variance, and an activation decay penalty. In contrast to a variational auto-encoder, we do not learn the variance, but rather keep it fixed. This prevents the possibility of the variance decreasing to a small value, ensuring that noise is applied equally on all components. Our objective function becomes:

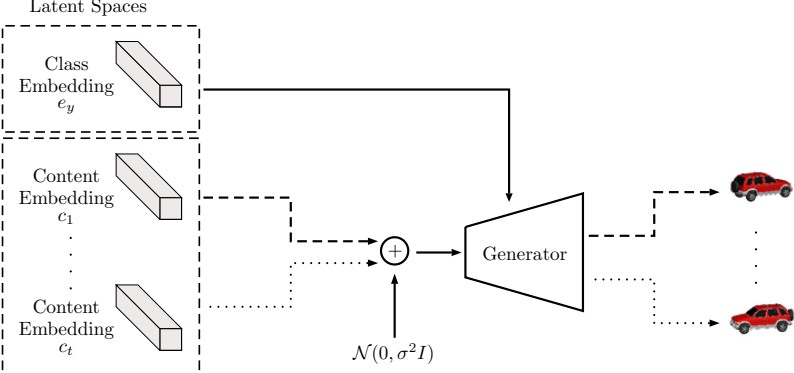

Figure 1: A sketch of the first stage: all class and content embeddings and the generator are jointly optimized. All images of the same class share a single class embedding. The content embeddings are regularized by a gaussian noise. By the end of this stage, the latent space of the training set is disentangled. Note that the second stage is not shown.

$$\mathcal{L} = \sum_{i=1}^{n} \|G_\theta(e_{y_i}, 0, c_i + z_i) - x_i\| + \lambda\|c_i\|^2 \quad z_i \sim \mathcal{N}(0, \sigma^2 I) \tag{6}$$

The first loss terms uses a VGG perceptual loss as implemented by Hoshen & Malik (2019). Unless stated otherwise, we optimize over class and content codes ($e_{y_i}$ and $c_i$) directly using latent optimization. All latent codes and the parameters of the generator are learned end-to-end using stochastic gradient descent:

$$\{e_1^*, .., e_k^*, c_1^*.., c_n^*, \theta^*\} = arg \min_{e,c,\theta} \mathcal{L} \tag{7}$$

### 3.2 Amortization for One-Shot Inference

Latent optimization, which is used effectively for training, requires optimization for every image (including at inference time). In the training set, a class embedding is shared across multiple images, which prevents the embedding from including content information. However, at inference time, a single image from an unknown class is observed. Optimizing over the latent codes for a single image leads to overfitting which results in entangled representations. Moreover, it requires iterative test-time inference since it does not perform amortized inference.

To this end, we introduce a second stage which learns class and content encoders that directly infer class $e_{y_i}$ and content $c_i$ representations from a single image $x_i$. The second stage effectively amortizes the results of the first stage and generalizes well to unseen classes and images. We train encoders $E_y : \mathcal{X} \rightarrow \mathcal{Y}$ and $E_c : \mathcal{X} \rightarrow \mathcal{C}$, which take as input an image $x_i$ and output its class and content embeddings that were learned by our method in the first stage. We also use a reconstruction loss, to ensure the representations learned in the second stage must reconstruct the original image $x_i$. The optimization objective is presented in Eq. 8. The optimization is over the parameters of encoders $E_y$ and $E_c$ (which are randomly initialized) and the parameters of the generator $G$ (which are initialized from the first stage). Note that the given $e_{y_i}$ and $c_i$ are the representations we have learned during the previous stage.

$$\mathcal{L}_E = \sum_{i=1}^{n} \|G_\theta(E_y(x_i), 0, E_c(x_i)) - x_i\| + \alpha_1 \cdot \|E_y(x_i) - e_{y_i}\|^2 + \alpha_2 \cdot \|E_c(x_i) - c_i\|^2 \tag{8}$$

After training, we can preserve the class of a new test image $\hat{x}_1$ and transfer over the content from another image $\hat{x}_2$ by decomposing the images into their disentangled class and content representa-

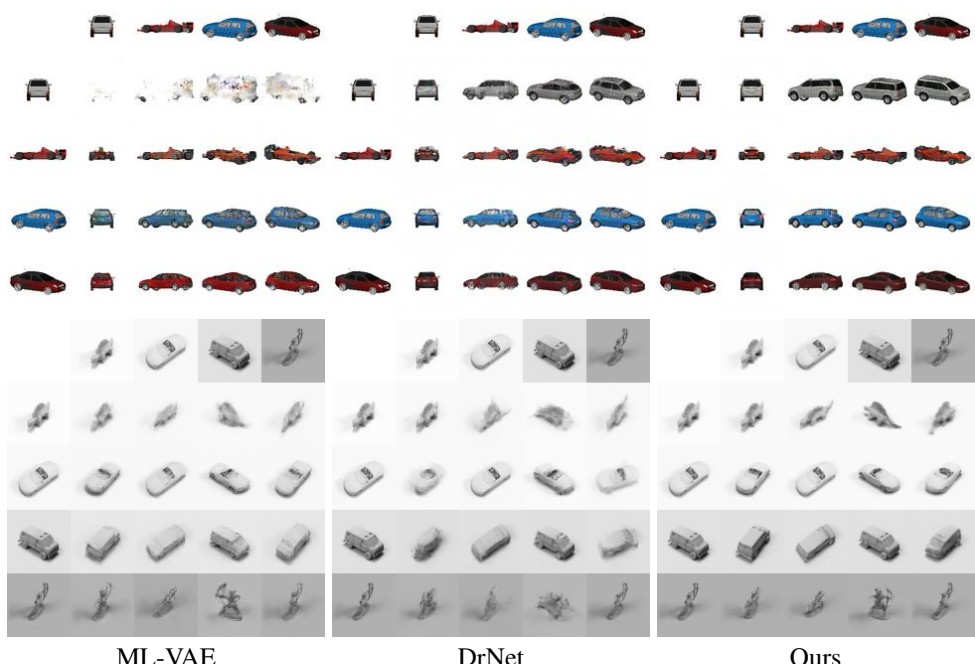

|  |  |  |
|:---:|:---:|:---:|
| ML-VAE | DrNet | Ours |

Figure 2: Comparison between our method and baselines on Cars3D (top) and SmallNorb (bottom).

tions and regenerating them as follows:

$$\hat{x}_{1 \leftarrow 2} = G_\theta(E_y(\hat{x_1}), 0, E_c(\hat{x_2})) \tag{9}$$

## 4 EXPERIMENTS

Our method is evaluated against SOTA techniques for class-supervised disentanglement. We do not compare to methods for fully-unsupervised disentanglement as the results are not directly comparable and their performance is inferior on the following benchmarks due to lower level of supervision. All the implementation details are provided in the Appendix A.1.

**Datasets:** We evaluate the performance of our method and the baselines on several datasets (each with the appropriate class labels): Cars3D (car model as class label, azimuth and elevation as content), SmallNorb (object type $\times$ lighting $\times$ elevation as class labels, azimuth as content), SmallNorb-Poses (object type $\times$ lighting as class labels, azimuth and elevation as content), CelebA (person identity as class label, other unlabeled transitory facial attributes e.g. head pose and expression as content), KTH (person identity as class label, other unlabeled transitory attributes e.g skeleton position as content), RaFD (facial expression as class label, rest as varied content). A more detailed description of each dataset and configuration can be found in the Appendix A.2.

**Baselines:** We compare our method against SOTA methods for class-supervised disentanglement. DrNet (Denton & Birodkar, 2017) and Szabó et al. (2018) encourage disentanglement by adversarial constraints, and ML-VAE (Bouchacourt et al., 2018) and Cycle-VAE (Harsh Jha et al., 2018) use variants of VAE equipped with grouped class accumulation and cycle constraints to discourage degenerate solutions. We also compare against StarGAN (Choi et al., 2018) in Multi-Domain translation. For fairness, we evaluate each baseline with $L_1$ and perceptual loss and report the best.

**Quantitative Experiments:**

*Content transfer experiments:* To test the quality of disentanglement, we measure the quality of content transfer in terms of perceptual similarity by LPIPS (Zhang et al., 2018). We use the content labels available in the Cars3D and SmallNorb datasets as ground truth for content transfer. For given test images $x_i$ and $x_j$, we measure the similarity between $x_{1 \leftarrow 2} = G_\theta(E_y(x_i), 0, E_c(x_j))$ and another image from the same class of $x_i$ matching the same content of $x_j$. For CelebA, given

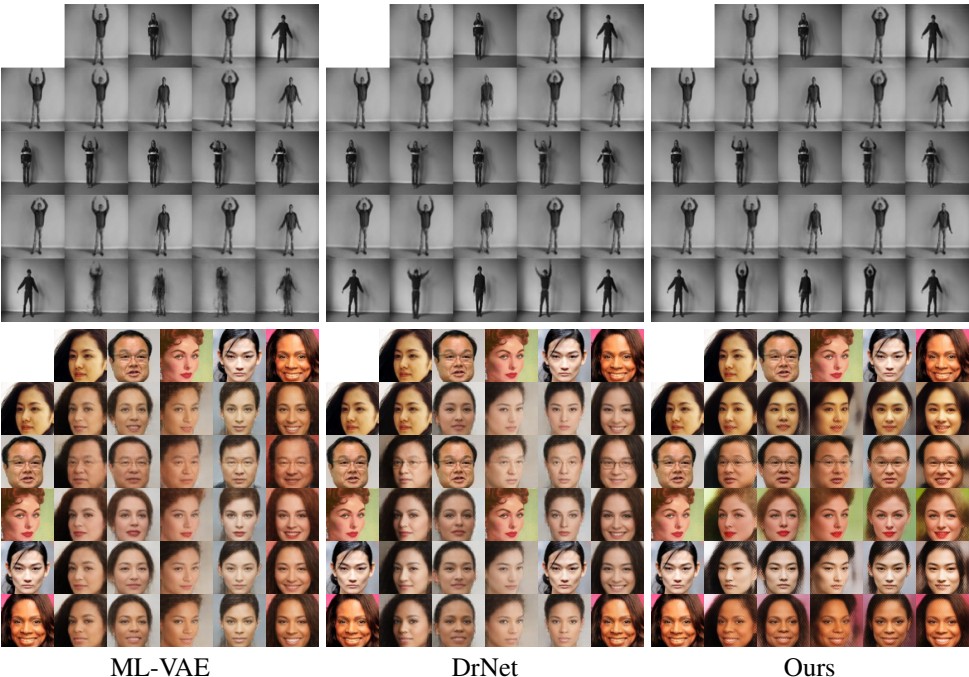

ML-VAE              DrNet              Ours

Figure 3: Comparison between our method and baselines on KTH (top) and CelebA (bottom).

two images of the same person, we infer the class (identity) representation from the first image and aim at reconstructing the second by extracting the content representation from an image of a different identity which has the most similar pose (nearest neighbour in the 68 facial-landmarks space). Results are reported in Tab. 1. It can be seen that we strongly outperform all the baselines.

*Classification experiments:* To assess the disentanglement of our learned representations, we follow the protocol in Harsh Jha et al. (2018) and train a classifier to classify class labels from content codes and vice versa. Results can be seen in Tab. 2. On all datasets, our model achieves near perfect disentanglement as the classifier could barely guess class labels from content codes by a random chance (same for the other direction). All the baselines fail to zero out the mutual information between the two representations. To conclude, our method is able to learn the most disentangled features without introducing adversarial constraints. For CelebA, in order to test if the content of an image is predictable from the class code we train a linear regression model to regress the position of 68 facial landmarks. It can be seen that the linear regression results in the highest error on our class representations, indicating the highest degree of disentanglement of our method. It should be noted that all methods could classify class labels from class codes and content labels from content codes very accurately (not shown).

*Facial expression transfer experiment*: We compare our method against StarGAN in the task of Multi-Domain translation. We follow the protocol in Choi et al. (2018) and compute the classification error of a facial expression classifier (trained on real images from RaFD) on synthesized images. We train both image translation models using the same training set and perform image translation on the same, unseen test set. As can be seen in Tab. 4, our model achieves lower classification error than StarGAN, indicating that our model produces more realistic facial expressions without using adversarial training.

**Qualitative Experiments:** We visually evaluate the results of our method against DrNet (strongest adversarial baseline) and ML-VAE (strongest non-adversarial baseline) in Fig. 2 and 3. In each experiment, we visualize switching between class (left column) and content (top row) codes for each pair within a set of 5 test images. On Cars3D, our method achieves excellent content transfer while keeping the class fixed. DrNet is mostly able to transfer the content, but it does not keep the car model fixed. ML-VAE results are of lower fidelity. On SmallNorb, our method works well, whereas the baseline methods struggle with preserving the identity of the object in some rotations (e.g. bottom row). On KTH both our method as well as DrNet perform well, although our method

Table 1: Content transfer reconstruction error (LPIPS ↓)

|  | Cars3D | SmallNorb | SmallNorb-Poses | CelebA |
|---|---|---|---|---|
| Szabó et al. (2018) | 0.137 | 0.417 | 0.214 | 0.331 |
| Cycle-VAE (Harsh Jha et al., 2018) | 0.141 | 0.197 | 0.202 | 0.228 |
| ML-VAE (Bouchacourt et al., 2018) | 0.132 | 0.210 | 0.173 | 0.222 |
| DrNet (Denton & Birodkar, 2017) | 0.095 | 0.166 | 0.152 | 0.229 |
| Ours | **0.078** | **0.117** | **0.106** | **0.197** |

Table 2: Classification accuracy of class labels from content codes ($y \leftarrow c$) and of content labels from class codes ($y \rightarrow c$) (lower indicates better disentanglement). Note that the last right column presents the error of face landmark regression from the class codes (higher is better).

|  | Cars3D | | SmallNorb | | CelebA | |
|---|---|---|---|---|---|---|
|  | $y \leftarrow c$ | $y \rightarrow c$ | $y \leftarrow c$ | $y \rightarrow c$ | $y \leftarrow c$ | $R(y) \rightarrow c$ |
| Szabó et al. (2018) | 0.91 | 0.82 | 0.36 | 0.37 | 0.09 | 3.59 |
| Cycle-VAE (Harsh Jha et al., 2018) | 0.08 | 0.80 | 0.27 | 0.79 | 0.14 | 3.14 |
| ML-VAE (Bouchacourt et al., 2018) | 0.77 | 0.96 | 0.90 | 0.93 | 0.17 | 3.98 |
| DrNet (Denton & Birodkar, 2017) | 0.26 | 0.68 | **< 0.01** | 0.78 | 0.03 | 3.23 |
| Ours | **< 0.01** | **0.01** | **< 0.01** | **0.05** | **< 0.01** | **4.75** |
| Random chance | < 0.01 | 0.01 | < 0.01 | 0.05 | < 0.01 | - |

achieves more accurate transfer (e.g. last image in the top row). ML-VAE fails to transfer the skeleton on some identities (e.g. last row). On CelebA, the baselines generally transfer head pose but do not preserve the person identity. Our method achieves better pose transfer than both baselines, and is able to maintain the identity.

**Non-adversarial unsupervised domain translation**: Our model can be used for the task of unsupervised domain translation by defining domain labels as class labels, as we demonstrate on RaFD dataset. We further extend our method for datasets in which classes (domains) exhibit in-class variations by introducing a preliminary step of clustering in-class styles (variations) into separate classes. For example, in the task of translating edge images into shoe images and vice-versa, we first form style clusters by applying k-means on style features extracted from first layer of a pretrained VGG model (**?**). The shoe images are therefore separated into sub-classes instead of a single class which contains high variation. This step decreases the degree of uncertainty in translating images between classes which exhibit in-class variations (See Appendix A.8 for more details). We then apply LORD treating the obtained style clusters as class labels on the Edges2Shoes dataset. Examples of translation diversity along with style-guided edges to shoe mapping are shown in Fig. 5. More examples of unsupervised domain translation are provided in Fig. 6 and Appendix A.9.

Table 3: An ablation study with several variants of LORD on Cars3D.

|  | Transfer error (LPIPS) ↓ | Classification accuracy ↓ | |
|---|---|---|---|
|  |  | $y \leftarrow c$ | $y \rightarrow c$ |
| Ours - amortized (w/ KL-divergence) | 0.094 | 0.95 | 0.96 |
| Ours - amortized (w/ Asymmetric noise) | 0.082 | 0.92 | 0.97 |
| Ours - semi amortized (w/ KL-divergence) | 0.095 | 0.93 | **0.01** |
| Ours - semi amortized (w/ Asymmetric noise) | 0.079 | 0.22 | **0.01** |
| Ours (w/o second stage) | 0.175 | 0.11 | 0.50 |
| Ours (w/o regularization) | 0.095 | 0.10 | **0.01** |
| Ours | **0.078** | **< 0.01** | **0.01** |

Table 4: Classification error (%)↓ on transferred facial expressions from RaFD.

| StarGAN (Choi et al., 2018) | Ours | Real Images |
|---|---|---|
| 2.2 | **1.8** | 0.8 |

| | Input | Angry | Contempt. | Disguste | Fearful | Happy | Sad | Surprised |
|---|---|---|---|---|---|---|---|---|
| Ours | | | | | | | | |
| StarGAN | | | | | | | | |

Figure 4: A qualitative comparison between our method (upper row) and StarGAN (bottom row) in facial expression transfer on RaFD. See Appendix A.5 for more results.

## 5 ABLATION ANALYSIS

We perform a careful ablation analysis on the components on our method, a summary of this study is presented in Tab. 3. Shared latent optimization vs. amortized inference: We train amortized variants of our model with feed-forward class and content encoders instead of optimizing over the latent codes directly. Class representations of samples from the same class are averaged within a mini-batch during training. It can be clearly observed from the results that class representations which are learned via amortized inference leak information about the actual content of each sample, resulting in entangled representations.

Moreover, we train semi-amortized variants of our model which leverages latent optimization for learning shared class representations, but uses a feed-forward encoder to infer the content code of an image. It can be noticed that this variant achieves sub-optimal performance as it leaks some class information into the content representation. In order to assess the inductive bias conferred by latent optimization, we measure the accuracy of classifying class labels from content codes after every epoch. The change in the amount of class-dependent information contained in the content codes is captured in Fig. 7. It can be observed that a randomly initialized content encoder (for amortization) encodes class-dependent information, which needs to be minimized as the training evolves. Initializing random content codes for latent optimization however provides no information about a specific class. By the end

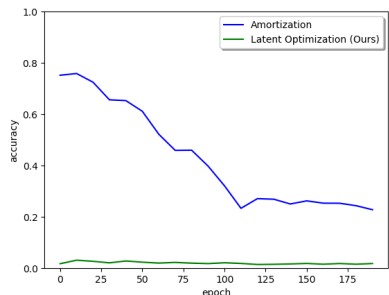

Figure 7: Accuracy of classifying class labels from content codes as evidence for the inductive bias conferred by latent optimization on Cars3D.

of training, amortized models often do not succeed in distilling the class-invariant information and provide entangled representations, while a model trained with latent optimization preserves a very

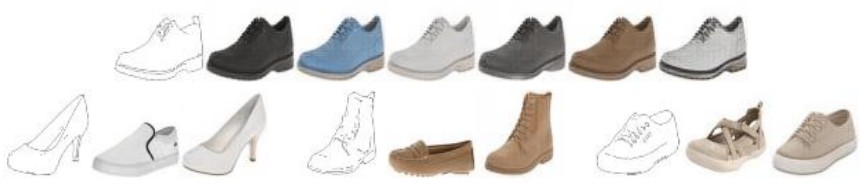

Figure 5: Examples of the diversity in translating edges to shoes (upper row) and style-guided translation (bottom row). Triplet order in bottom row (left to right): edges, style, translation.

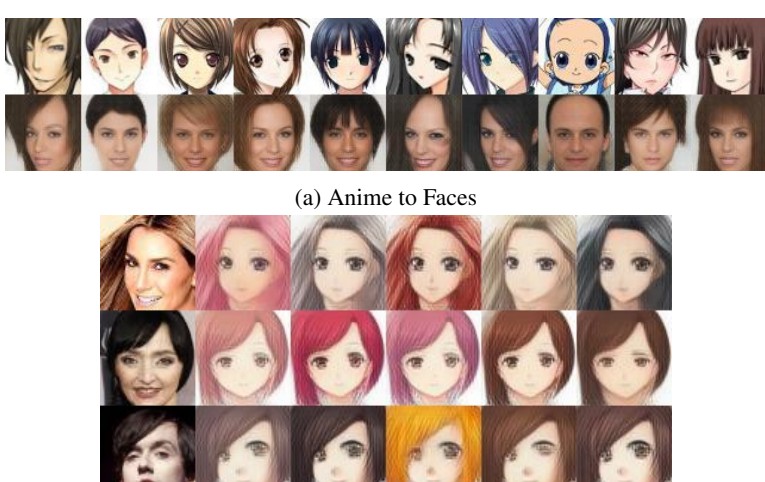

(a) Anime to Faces

(b) Diversity in translating Faces to Anime (using style clustering).

Figure 6: Examples of translations between anime and faces from CelebA using our method.

high degree of disentanglement. We hypothesize that achieving similar degree of disentanglement by amortization requires a more sophisticated objective and a more careful hyperparameter tuning. An extended study is presented in Appendix A.4. It should be noted that latent optimization requires more iterations than optimizing an amortized encoder and leads to a slower convergence (the number of iterations increased by $\times 2$ in our experiments). In both the amortized and semi-amortized models, we find that the KL-divergence fails to regularize the information leakage from the class representation into the content representations. A visualization of the partial posterior collapse can be found in the Appendix A.3. We finally demonstrate the importance of our second stage by assessing the performance after the first stage only. This can be done by optimizing over the latent codes of a new test image while keeping the rest of the model frozen. As can be seen, this approach suffers from low performance in all metrics. The effect of the asymmetric noise regularization can be observed from the inferior performance of training our model without regularization. A qualitative visualization of this analysis is provided in Appendix A.6.

## 6 DISCUSSION

*Non-Adversarial training:* Differently from most other previous works, we do not use adversarial training to enforce disentanglement between the class and content. Non-adversarial training has significant advantages in the ease of optimization. Interestingly, we achieve state-of-the-art performance without any adversarial constraints. We believe this should motivate researchers to further develop non-adversarial approaches.

*Perceptual loss:* For training our model, we use a perceptual loss, originally trained on the imagenet dataset. This is not extra supervision, as the imagenet dataset is not strongly related to any of the tested datasets. In our experiments we found the perceptual loss was helpful to other method that did not use GANs on the output image (even if they used GANs on the intermediate features representations e.g. Denton & Birodkar (2017)). In line with other work Hoshen & Malik (2019), we found that perceptual losses are very helpful for latent optimization.

## 7 CONCLUSION

We present an effective approach for class-supervised image disentanglement, using shared latent optimization, an asymmetric regularization and a second amortization stage for single-shot generalization. Our approach achieves state-of-the-art performance compared to both adversarial and non-adversarial disentanglement methods. We finally show how style clustering can extend our method for tackling domain translation as an inter-class disentanglement with promising results.

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

# A    APPENDIX

## A.1    IMPLEMENTATION DETAILS

The architecture of the generator consists of 3 fully-connected layers followed by 6 convolutional layers (the first 4 of them are preceded by an upsampling layer and followed by AdaIN normalization). We set the size of the content latent code to 128 and the size of the class code to 256 in all our experiments. We regularize the content embeddings with an additive gaussian noise with $\mu = 0$ and $\sigma = 1$ and an activation decay with $\lambda = 0.001$. We perform the latent optimization using SGD utilizing the ADAM method for 200 epochs, with learning rate of 0.0001 for the generator and 0.001 for the latent codes. For each mini-batch, we update the parameters of the generator and the latent codes with a single gradient step each. For the second stage, the class and content encoders are CNNs with 5 convolutional layers and 3 fully-connected layers.

## A.2    DATASETS

*Cars3D (Reed et al., 2015)*: This dataset consists of 183 car CAD models, each rendered from equispaced 24 azimuth directions and 4 elevations. We define the car model as the class and the rest as content. We use 163 car models for training and the other 20 are held out for testing.

*SmallNorb (LeCun et al., 2004)*: This dataset contains images of 50 toys belonging to 5 generic categories: four-legged animals, human figures, airplanes, trucks, and cars. The objects were imaged by two cameras under 6 lighting conditions, 9 elevations (30 to 70 degrees every 5 degrees), and 18 azimuths (0 to 340 every 20 degrees). We use this dataset in two configurations: i) *SmallNorb*: 25 separate identities for training and 25 for testing, treating lighting and elevations as part of the object class, and azimuth as the varied content. This configuration is used for evaluating the generalization capability of the disentanglement methods from a very limited set of seen classes. ii) *SmallNorb-Poses*: Using all the classes for training, holding out 10% of the images for testing. In this case we treat the elevation as part of the varied content as well.

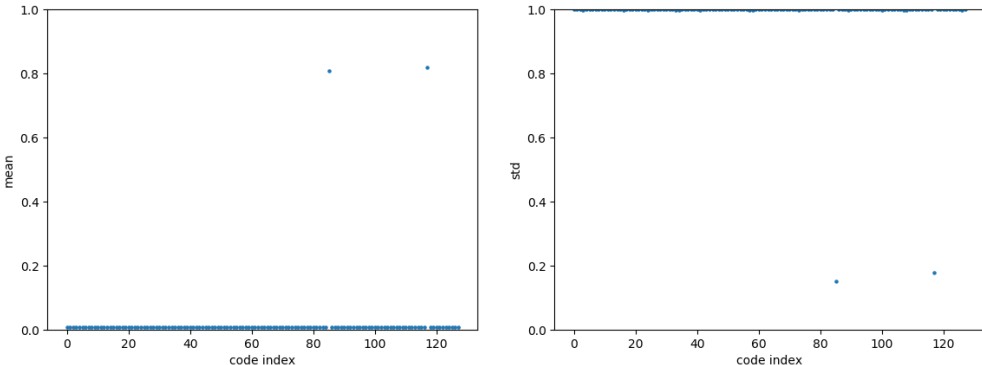

Figure 8: Evidence for a partial posterior collapse with KL-divergence. 126 out of 128 components of the content code collapse to match a perfect standard normal distribution with zero mean and a unit standard deviation. The remaining two components sustain much higher mean and much lower standard deviation. This prevents the regularization from acting as a tight bottleneck.

*CelebA (Liu et al., 2015)*: CelebA contains 202,599 facial images of 10,177 celebrities. The faces are aligned and cropped to contain only the facial region. We designate the person identity as the class, and transitory facial attributes such as head pose and expression as content. 9,177 classes are used for training and the other 1,000 are held out for testing.

*KTH (Laptev et al., 2004)*: KTH contains videos of 25 people, performing 6 different activities in different settings. We designate person identity as class, and transitory attributes (predominantly skeleton position) as content. Due to the very limited amount of subjects, we use all the identities for training, holding out 10% of the images for testing.

*RaFD (Langner et al., 2010)*: RaFD consists of 4,824 images collected from 67 participants making eight facial expressions in three different gaze directions, which are captured from three different angles. We treat the facial expression as class and rest as varied content, holding out 10% of the images for testing.

*Edges2Shoes (Yu & Grauman, 2014)*: A collection of 50,000 shoe images and their edge maps.

*Anime (Mckinsey, 2019)*: A dataset consisting of 63,632 anime faces.

In all the experiments, images are resized to 64x64 resolution to fit the same architecture in LORD and the baselines. For evaluation on RaFD we follow the protocol in StarGAN (Choi et al., 2018) and crop the images to 128x128.

### A.3    KL-DIVERGENCE POSTERIOR COLLAPSE

We provide evidence for the partial posterior collapse we experienced when regularizing the content codes with KL-divergence. Fig. 8 shows the mean and standard deviations of each of the 128 components of the content code (averaged over all samples in the dataset) in a model trained on SmallNorb. It can be seen that 126 out of 128 components of the content code collapse to match a perfect standard normal distribution, while in the remaining 2 components the standard deviation is reduced dramatically along with a substantial increase in the mean. This phenomenon implies that regularizing the distribution of the content codes with KL-divergence may require additional attention and a careful hyperparameter tuning. We find in our experiments that the asymmetric regularization introduced in our method results in better disentanglement.

### A.4    INDUCTIVE BIAS OF LATENT OPTIMIZATION

We further provide the train and test losses (along with their decomposition into reconstruction and regularization terms) for the different variants of LORD in Tab. 5. It can be seen that the semi-amortized model (with asymmetric noise) achieves a slightly lower reconstruction loss and a lower

Table 5: A summary of train / test losses of the different variants of LORD on Cars3D. Regularization measures the activation penalty of the content codes.

|  | Reconstruction | Regularization | Total |
| --- | --- | --- | --- |
| Ours - amortized (w/ KL-divergence) | 40.53 / 50.90 | 3.13 / 3.20 | 40.54 / 50.90 |
| Ours - amortized (w/ Asymmetric noise) | 26.62 / 42.02 | 57.80 / 57.25 | 26.68 / 42.08 |
| Ours - semi amortized (w/ KL-divergence) | 16.07 / 45.41 | 10.24 / 9.23 | 16.08 / 45.42 |
| Ours - semi amortized (w/ Asymmetric noise) | 13.29 / 44.39 | 55.47 / 51.95 | 13.34 / 44.44 |
| Ours | 13.88 / 44.78 | 56.17 / 54.65 | 13.94 / 44.83 |

activation penalty (regularization of the content codes) than our latent optimization (fully unamortized) model. This emphasizes the effect of the inductive bias conferred by latent optimization which despite the higher losses results in a better disentanglement performance, as presented in Tab. 3 and Fig. 7. The second semi-amortized model, regularized with KL-divergence, achieves a much lower activation penalty in the content codes as it collapses almost all means to zero. In all the experiments we set $\lambda = 0.001$ (Eq. 6). In CelebA and SmallNORB, we failed to achieve better optima in both reconstruction and regularization losses, when using the semi-amortized model compared to our fully-unamortized model: [CelebA] ours: Rec = 100.38, Reg = 42.99 — semi-amortized ($\lambda = 0.001$): Rec = 82.07, Reg = 128.84 — semi-amortized ($\lambda = 0.01$): Rec = 104.23, Reg = 8.54. [SmallNORB] ours: Rec = 32.09, Reg = 16.54 — semi-amortized ($\lambda = 0.001$): Rec = 31.15, Reg = 17.83. This behaviour emphasizes the difficulties in balancing the objectives for encouraging disentanglement using amortized inference.

## A.5 QUALITATIVE RESULTS

We provide more qualitative results in Fig. 9, 10, 11, 12 and 13.

## A.6 QUALITATIVE COMPARISON TO BASELINES

In addition to the quantitative assessment presented in the ablation study, we provide a qualitative comparison on CelebA of our method and 3 of our baselines in Fig. 14.

## A.7 INTRA-CLASS VARIATION

Our image formation model, models images as being formed by class, content and residual (style) codes. The intra-class variation is formed by both the content and the residual information. The content is transferable between classes, the residual information is not. Given class and content codes, if the residual information is small, reconstruction will be successful (as demonstrated in our experiments). If the residual information is very significant, it will not be possible to reconstruct images well only based on class and content leading to poor image formation models. For example, in the Cars3D experiment, the class labels represent the car model, content codes represent azimuth and elevation, and there is no residual information. In this case LORD performs well. We perform an exploratory experiment in which we aggregate similar car models into a single unified class (163 original car models are clustered into 50 super classes). In this case, the residual information contains the specification of the exact car model within the super class. The residual information is therefore significantly larger. The class and content information is not sufficient for reconstructing the original image perfectly. We demonstrate the degradation in the reconstruction in Fig. 15.

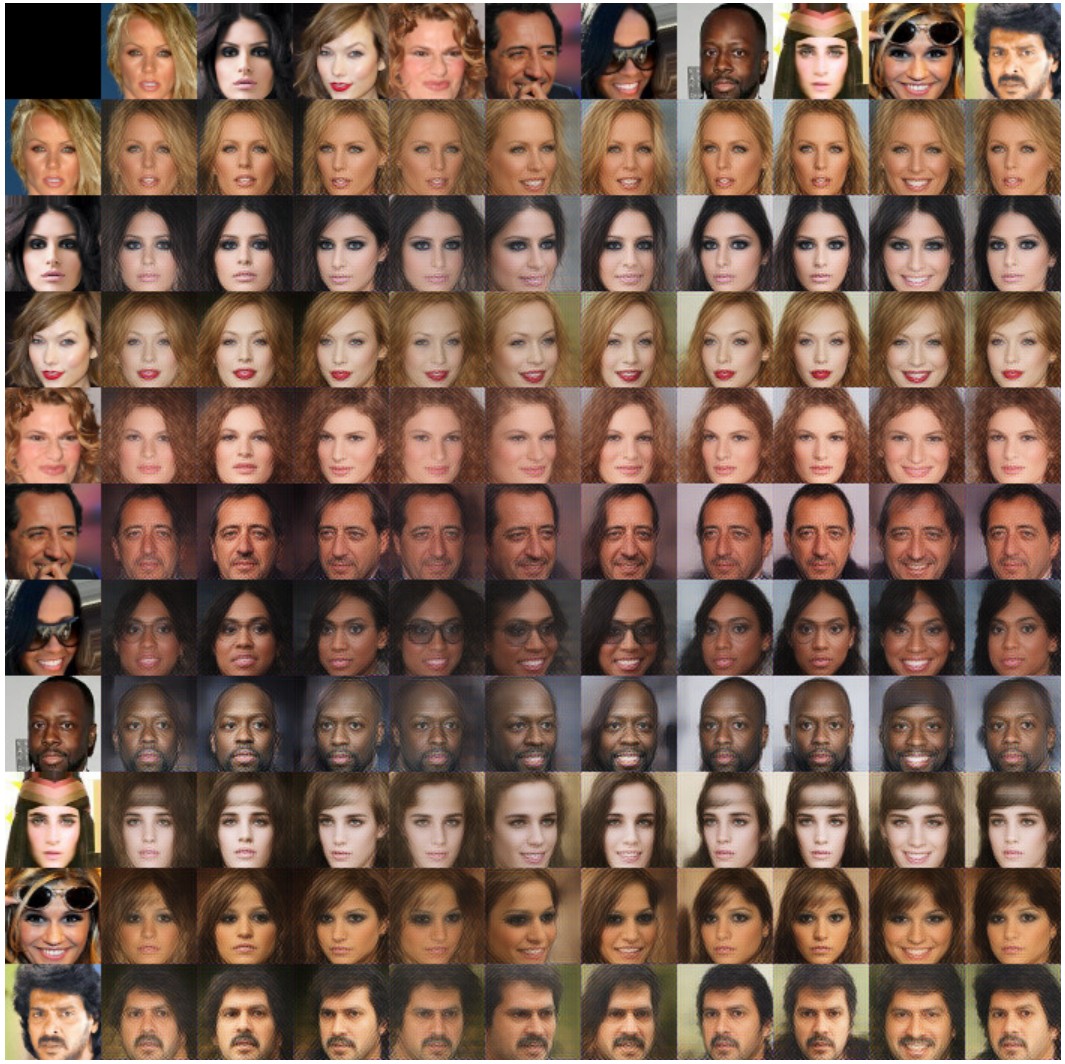

Figure 9: More qualitative results of our method in transferring content between classes on CelebA.

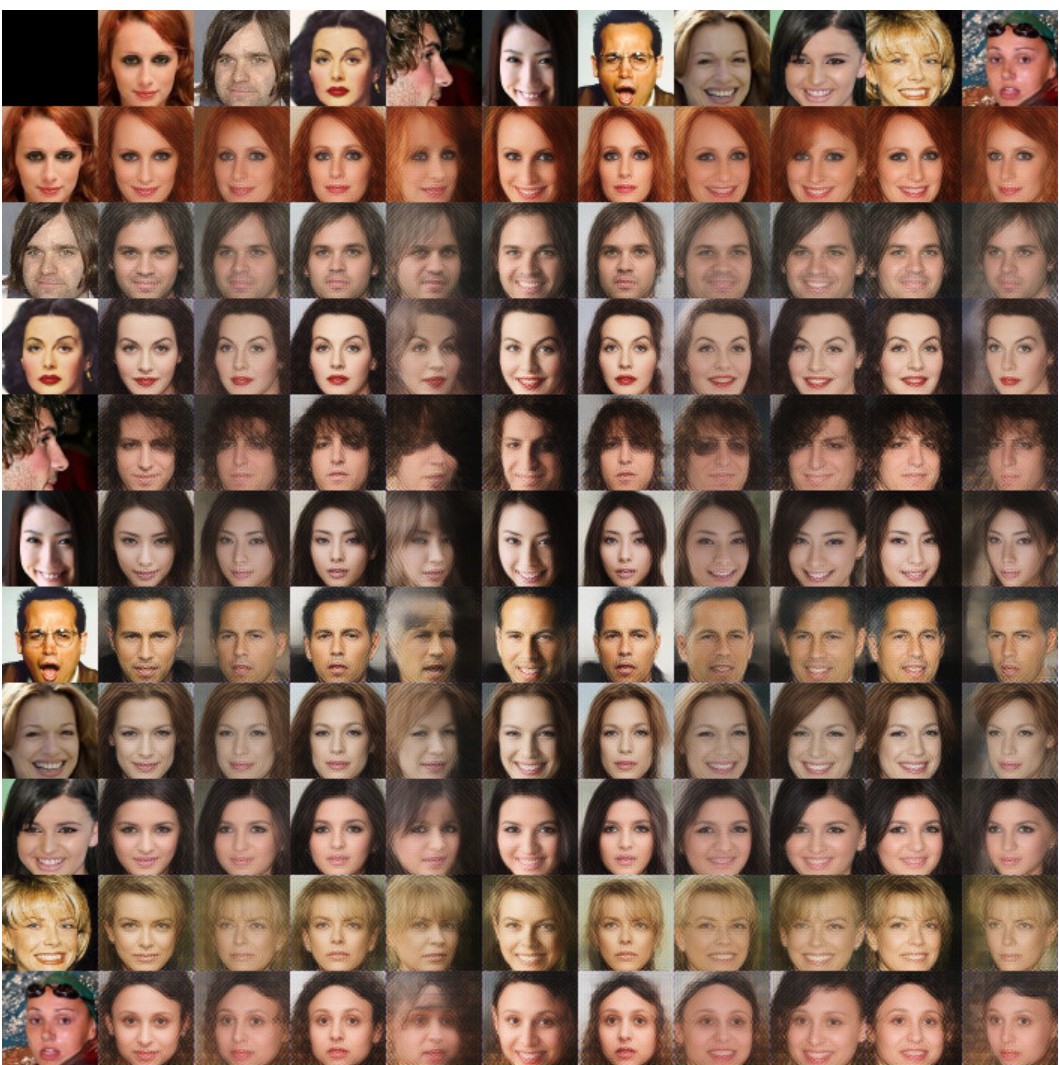

Figure 10: More qualitative results of our method in transferring content between classes on CelebA.

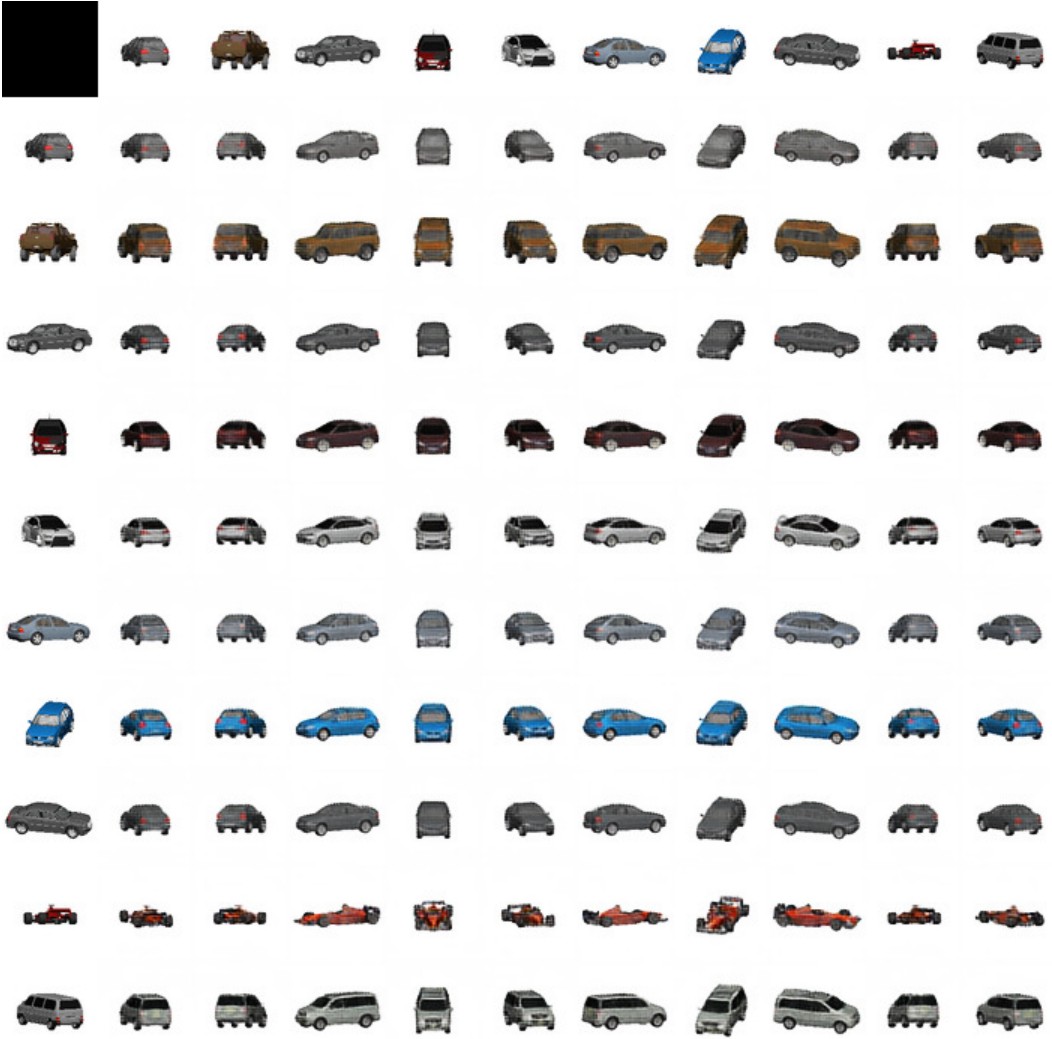

Figure 11: More qualitative results of our method in transferring content between classes on Cars3D.

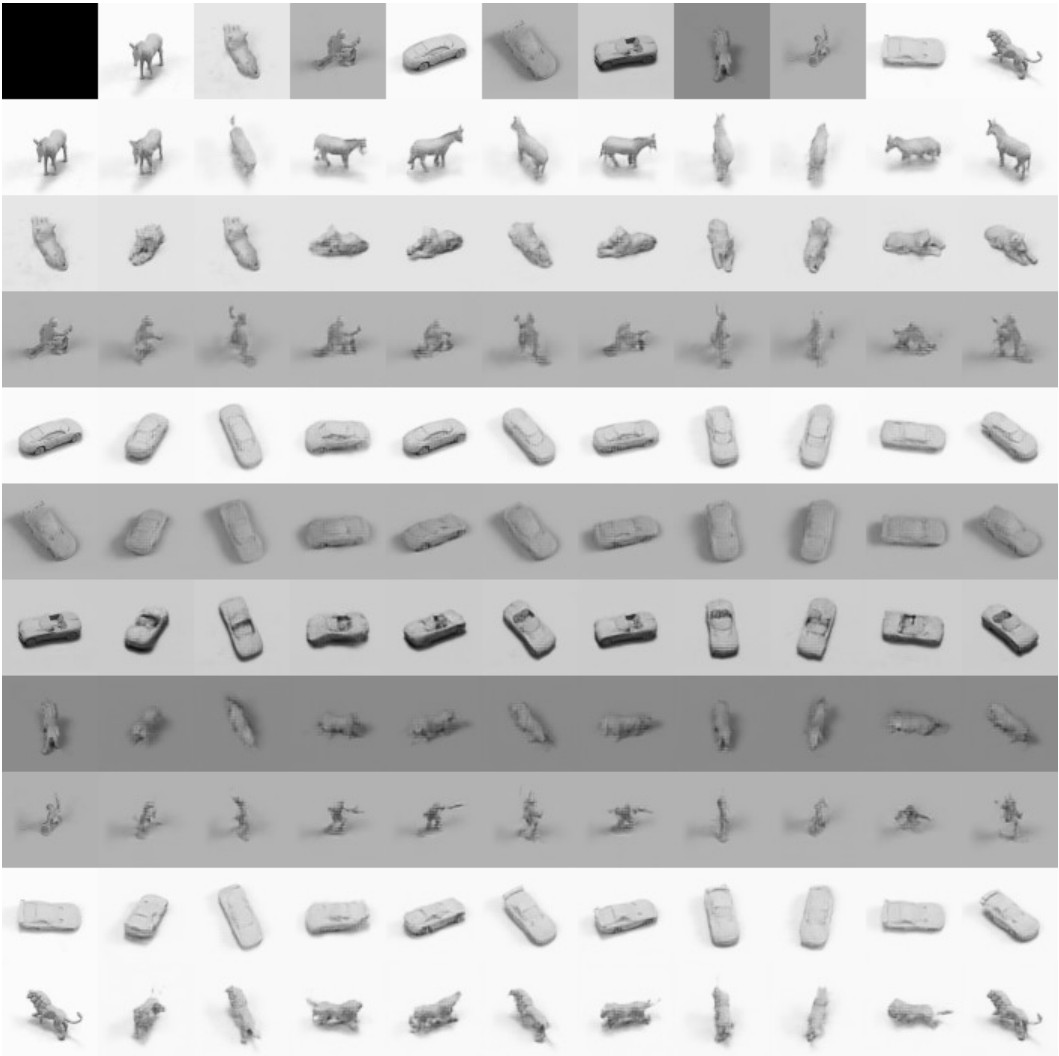

Figure 12: More qualitative results of our method in transferring content between classes on Small-Norb.

| Input | Angry | Contempt. | Disguste | Fearful | Happy | Sad | Surprised |
|---|---|---|---|---|---|---|---|

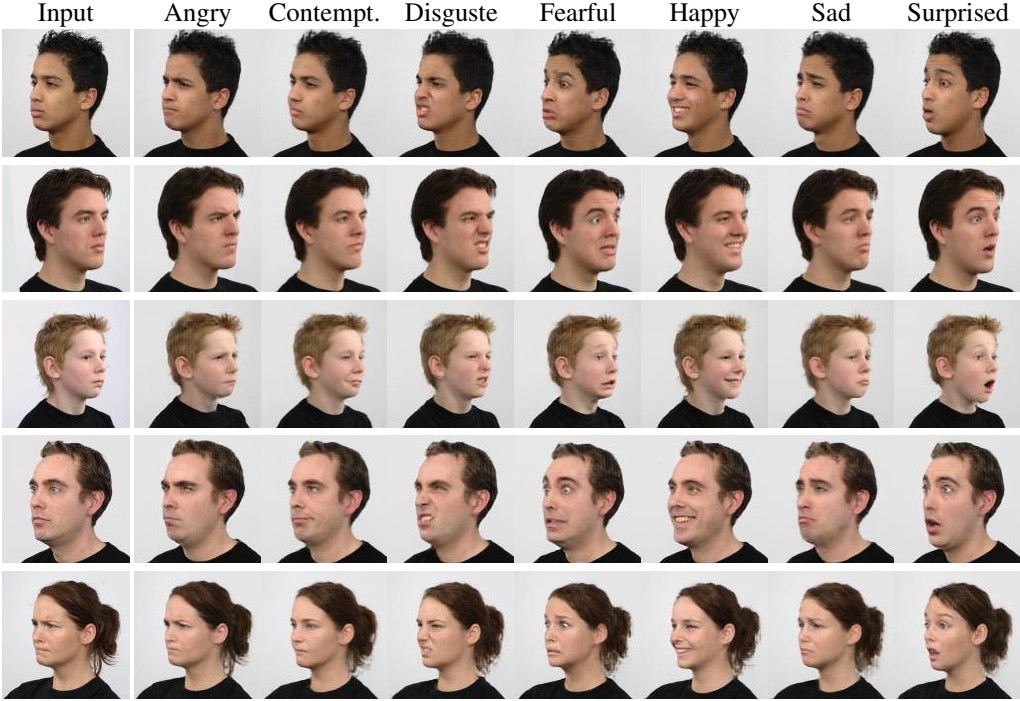

Figure 13: More qualitative results of our method in facial expression transfer on RaFD.

### A.8 STYLE CLUSTERING

We describe the preliminary step of per-class style clustering in Alg. 1.

---
**Algorithm 1:** Style clustering

---
> **Input:** $n$ images $x_1, x_2, ..., x_n \in \mathcal{X}$ and respective class labels $y_i \in [k]$
>   Number of styles per class $l \in \mathcal{N}$
>   Feature extraction function $\phi : \mathcal{X} \to \mathcal{R}^d$
> **Output:** Per class style labels $\psi : [n] \to [k] \times [l]$
> $\forall i \in [n], f_i \leftarrow \phi(x_i)$       `// extract features from images`
> $\forall j \in [k], t_i \leftarrow \text{k-means}_l(\{f_i | y_i = j\})$    `// cluster class` $j$ `into` $l$ `styles`
> $\forall i \in [n], \psi(i) \leftarrow (y_i, t_i)$      `// assign joint class and style labels`

---

We provide samples in Fig. 16 of clustering shoe images with k-means ($k = 2, l = 100$) using style features (Li et al., 2017) extracted from a pretrained VGG model.

### A.9 DOMAIN TRANSLATION RESULTS

We extend our unsupervised domain translation approach to support multi-attribute class labeling. We use the 40 annotated attributes in CelebA and use them to supervise 40 different shared class embeddings in addition to a single content embedding optimized per image. We demonstrate the effectiveness of this extended LORD approach in translating males to females and vice versa in Fig. 17.

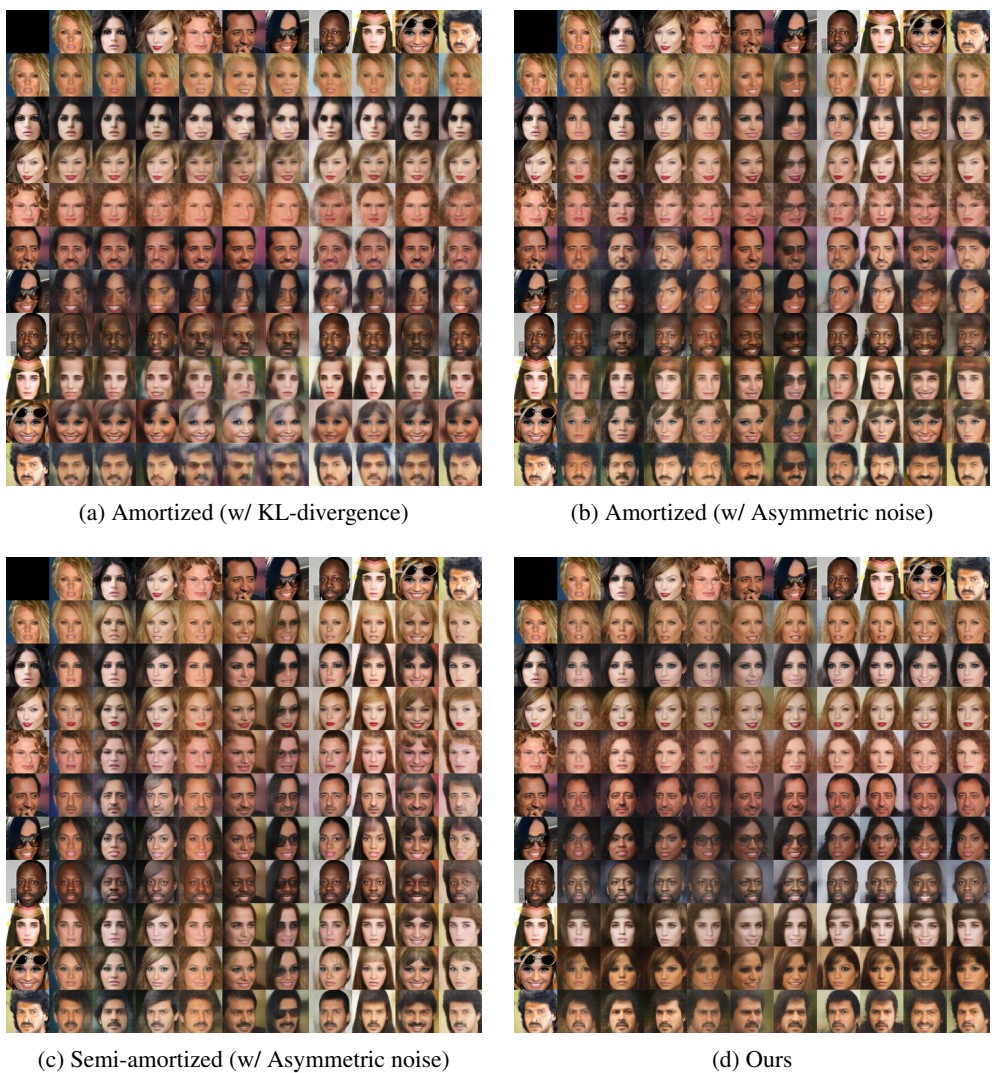

(a) Amortized (w/ KL-divergence)

(b) Amortized (w/ Asymmetric noise)

(c) Semi-amortized (w/ Asymmetric noise)

(d) Ours

Figure 14: Qualitative comparison on CelebA between our method and 3 of our ablation base-lines. Fully-amortized models (a, b) fail to preserve the class (person identity) across different content codes and introduce several artifacts, showing their lower degree of disentanglement. Semi-amortized model regularized with asymmetric noise (c) transfers over unreliable properties between identities (such as hair style). Our model (d) learns to disentangle the intrinsic characteristics of each identity and provides the best disentanglement and highest quality.

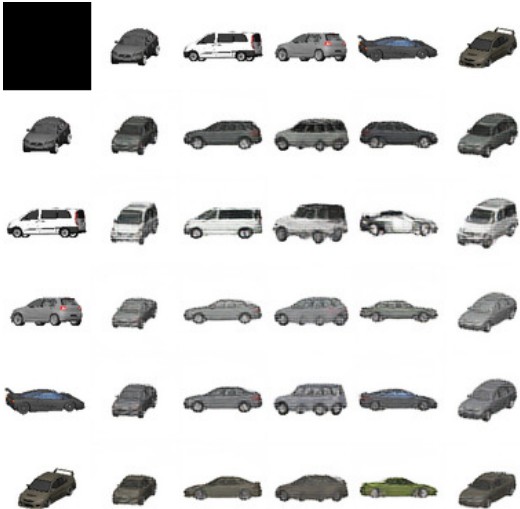

Figure 15: A visualization of the degradation in reconstruction and disentanglement quality in cases where classes exhibit intra-class variations. It can be observed that the car model is not preserved well across different content codes.

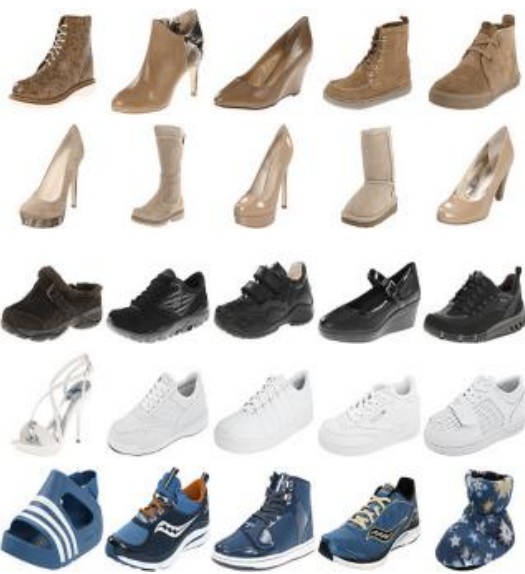

Figure 16: Random samples from clusters of shoe images formed by k-means on style features extracted from a pretrained VGG model.

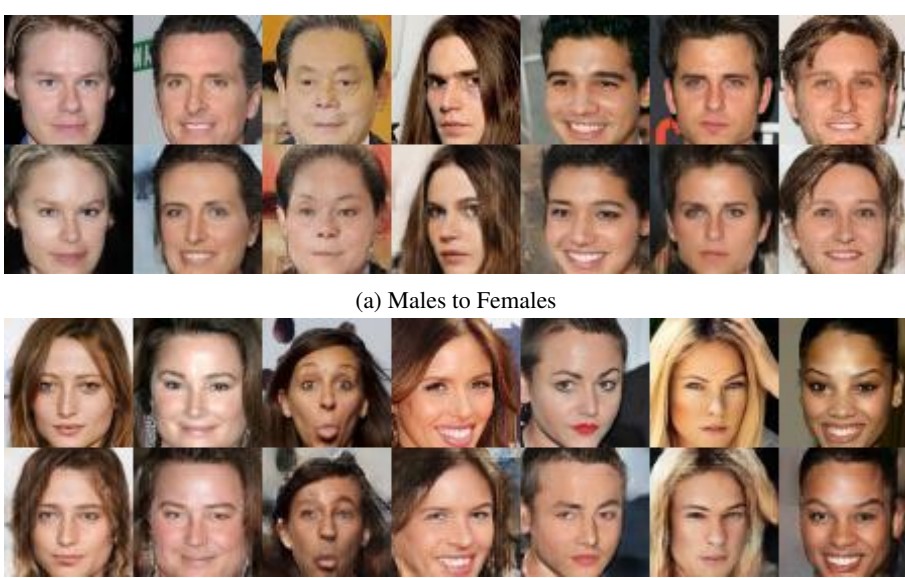

(a) Males to Females

(b) Females to Males

Figure 17: Examples of translations between genders of faces from CelebA using our method.

