# OpenReview forum: "Demystifying Inter-Class Disentanglement"
_ICLR.cc/2020/Conference — Accept (Poster)_

### Official Review · AnonReviewer3 · 2019-10-20
**Official Blind Review #3**

**Rating:** 6

**Review:**

The paper proposes a framework, called LORD, for better disentanglement of the class and content information in the latent space of image representations. The authors operate in a setting where the class labels are known. The authors perform
optimization in the latent space to obtain these representations and argue that this is simpler and effective than adversarial and amortized inference approaches.

The main issue that the authors tackle is the information leakage between the representations for the class and content. The authors suggest several fixes for this. Firstly, the paper makes a distinction between content and style. Content is defined as the information that is unchanged when only the class labels are changes in the data generative process. The inherent randomness in the data generative process is defined as the style.

To disallow, the leakage from content/style code to class code, the authors suggest learning fixed codes for each class that does not vary across images. That is if two images have the same class by virtue of design they will have the same class codes.

The reverse, leakage from class codes to content codes is achieved by adding any asymmetric noise regularization term. This also seems to be aimed at reducing the total variability in the content codes. The authors claim that this is better than the bottleneck approach such as matching the code distribution to uniform prior and provide empirical evidence. Though in theory, it is not clear why one is better than the other. How was the sigma tuned for the regularization? Are the results dependent on this parameter?

After learning the class and content embeddings for each sample in the training example, a pair of encoders are learned to predict these codes for unseen test images, without the need for optimization.

Other comments:

The style code being 0 is not clear. Does the randomness in content code during the training account for the variations in the images not covered by class code and content code.

The methods seem heavily reliant on the imagenet trained VGG perceptual loss. This does not seem to be an issue in the datasets shown, do the authors anticipate any limitations generalizing to datasets such as in medical domains, etc.

Why is lighting chosen as the class label in the datasets? It will be interesting to see how the results change with different subsets of class labels and what is captured in the style codes.

What are limitations from the assumption of low variability within a class?

Typo: Page 4 - minimally -> minimality






**Experience Assessment:**

I have published in this field for several years.

**Review Assessment: Checking Correctness Of Derivations And Theory:**

I assessed the sensibility of the derivations and theory.

**Review Assessment: Checking Correctness Of Experiments:**

I assessed the sensibility of the experiments.

**Review Assessment: Thoroughness In Paper Reading:**

I read the paper at least twice and used my best judgement in assessing the paper.

---

> ### Author Response · Authors · 2019-11-10
> **Response to Review #3**
>
> We thank the reviewer for the dedicated and positive review.
>
> “How was the sigma tuned for the regularization? Are the results dependent on this parameter?”: In all our experiments, we used a fixed value of sigma=1, it is possible that better results may be obtained with a different value of sigma.
>
> “Does the randomness in content code during the training account for the variations in the images not covered by class code and content code.”: The purpose of regularizing the content code with random noise and activation decay is to obtain disentangled representations (by minimality of information) and not for diverse generation. In the datasets considered, the assumption that variation not covered by class and content is small holds quite well. In cases where this assumption is insufficient, we introduced the preliminary style clustering step.
>
> “reliant on the imagenet trained VGG perceptual loss … do the authors anticipate any limitations generalizing to datasets such as in medical domains, etc.”:  Relying on the VGG loss is obviously a limitation for non-image datasets (although perceptual losses exist in other modalities). Regarding images, previous research has shown that the VGG loss is quite generally effective e.g. Yang et al. [1] used the VGG loss successful in the medical domain.
>
> “Why is lighting chosen as the class label in the datasets?”: We followed the SmallNORB protocol in DrNet, which kept object identity, lighting and elevation constant. We further extended this protocol to another configuration in which the elevation also varied.
>
> “What are limitations from the assumption of low variability within a class?”: The limitation is in the case where high intra-class variability is not explained by the content (and is not expected to be transferred across classes). In cases where the assumption is not satisfied, reconstruction will suffer. Our method can work even in such cases e.g. in the CelebA experiments. In cases where the non-content intra-class variability is very high, we perform the preliminary style-clustering step (e.g. Edges2Shoes). For more details, please see our response to R1.
>
> [1] Yang, Qingsong, et al. "Low-dose CT image denoising using a generative adversarial network with Wasserstein distance and perceptual loss." IEEE transactions on medical imaging, 2018.

---

### Official Review · AnonReviewer1 · 2019-10-21
**Official Blind Review #1**

**Rating:** 6

**Review:**

This paper proposes LORD, a novel non-adversarial method of class-supervised representation disentanglement. Based on the assumption that inter-class variation is larger than intra-class variation, LORD decomposes image representation into two parts: class and content representations, with the purpose of getting disentangled representation in those parts.
Inspired by ML-VAE, authors try to: 1. learn to reconstruct the original image of disentangled representation. 2. eliminate the class information from content code by asymmetric noise regularization. The experimental results indicate that LORD succeeds to disentangle class information on content codes, while it outperforms style-content disentangled representations on style switching tasks (Figure 2 & 3).
Strengths:
1.LORD achieves significantly better performance than the state-of-the-art baseline on non-adversarial disentanglement methods.
2., In terms of confusing the classifier in “Classification experiments” (Table2), disentangled content representation of LORD behaves like a random guess. This shows that LORD is indeed in preventing class information from leaking into content representations.
Weaknesses:
1. This paper is based on the assumption that “inter-class variation is larger than intra-class variation”. Authors should verify their assumption by quantitative results and illustrate the importance of inter/intra-class variation (e.g. how much information we may lose if ignoring the intra-class variation).
2. Authors claim that no information leakage between class and content representation in Sec 1.1. However, the experiments only verify “no class information in content code”, but miss the inverse proposition (Is there any content information is class code?)


**Experience Assessment:**

I have read many papers in this area.

**Review Assessment: Checking Correctness Of Derivations And Theory:**

I assessed the sensibility of the derivations and theory.

**Review Assessment: Checking Correctness Of Experiments:**

I assessed the sensibility of the experiments.

**Review Assessment: Thoroughness In Paper Reading:**

I read the paper thoroughly.

---

> ### Author Response · Authors · 2019-11-10
> **Response to Review #1**
>
> We thank the reviewer for the dedicated and positive review, and for recognizing the novelty of the method and significance of the results.
>
> “LORD achieves significantly better performance than the state-of-the-art baseline on non-adversarial disentanglement methods”: We wish to highlight that additionally to outperforming non-adversarial methods, our method outperforms state-of-the-art adversarial methods such as DrNet and StarGAN.
>
>  “Authors claim that no information leakage between class and content representation … experiments only verify 'no class information in content code', but miss the inverse proposition”: Table.2 already contains both class classification from content code as well as content classification from class code. The results show that our method achieves near perfect disentanglement on both directions.
>
> “This paper is based on the assumption that ‘inter-class variation is larger than intra-class variation’. Authors should verify their assumption”: Our image formation model, models images as being formed by class, content and residual (style) codes. The intra-class variation is formed by both the content and the residual information. The content is transferable between classes, the residual information is not. Given class and content codes, if the residual information is small, reconstruction will be successful (as demonstrated in our experiments). If the residual information is very significant, it will not be possible to reconstruct images well only based on class and content leading to poor image formation models. For example, in the Cars3D experiment, the class labels represent the car model, content codes represent azimuth and elevation, and there is no residual information. In this case LORD performs well. We perform an exploratory experiment in which we aggregate similar car models into a single unified class (163 original car models are clustered into 50 super classes). In this case, the residual information contains the specification of the exact car model within the super class. The residual information is therefore significantly larger. The class and content information is not sufficient for reconstructing the original image perfectly. Quantitatively, we found that the reconstruction error increased from 32.5 to 55.64. A visualization of this experiment is provided in the appendix (A.7). It should be noted that our method can work well in cases where there is a moderate amount of residual information, e.g. in the CelebA experiments. Moreover, in datasets which exhibit large intra-class variations (e.g. Edges2Shoes) we introduce our preliminary step of style clustering which significantly reduces intra-class variation.

---

### Official Review · AnonReviewer2 · 2019-10-25
**Official Blind Review #2**

**Rating:** 6

**Review:**

Summary: this paper proposes to basically combine class-conditional noisy autoencoding with GLO to achieve disentanglement while having only access to the content labels. They demonstrate that the method achieves impressive empirical results both in terms of disentanglement and a limited experiment on unsupervised domain translation.

Decision: Weak Reject. I find the experimental results in this paper very appealing. However, I am weakly rejecting the paper because of 1) some problematic claims put forth in the paper, which I worry might mislead the reader and 2) lack of clarity in describing the procedure in the unsupervised domain translation setting.

Here are some main comments:

1.  KL-divergence v. asymmetric noise
First, the authors claim that regularizing with KL-divergence leads to posterior collapse. But the particular experimental set up is tested on SmallNORB, which only has a small handful of factors of variation anyway). That KL-divergence “causes” posterior collapse is a claim that must be made very carefully. There are some very specific conditions under which this is known to be true empirically (for example, see the experiments in Burda’s IWAE paper and Hoffman’s DLGM paper), but in general, one should be careful with this claim. Can the authors please walk back on this statement?

Second, it is worth noting that asymmetric noise regularization is itself actually a special case of KL-divergence regularzation. When q(z|x) is forced to have a globally fixed variance, KL-divergence regularization becomes asymmetric noise regularization.

2. Cost of training
One thing I feel should be made more clear in the paper is the training cost of GLO v. amortized models. How much slower is GLO compared to amortized models? How many iterations do you employ on a given minibatch of data when using GLO?

3. Ablation study
First, I think the authors should show us the actual visualizations for the amortized models. Without visual inspection, it’s hard to gauge the significance of the numbers in Table 3.

Second, the authors observed that the amortized models leak class information into the content representation. I find it fascinating that GLO does not. I would like the authors to dig deeper into what exactly is the inductive bias conferred by latent optimization. As of the moment, claim that “this variant is inferior to our fully unamortized model as a result of an inductive bias conferred by latent optimization” is a vacuously true statement since we know that amortized models and unamortized models should in theory have equivalent behavior in the infinite-capacity / oracle optimizer regime. I request that the authors show us the training and test losses (Eq 6 and its decomposition into reconstruction + regularization terms). Inspecting it may shed light on the inductive bias.

4. Unsupervised Domain Translation
The result looks very good. However, the experimentation is too limited. I recommend that the authors try at least one other dataset.

Furthermore, the description of how to apply LORD to unsupervised domain translation is uncomfortably vague. I am not sure if the provided code and description in the main text allows for reproduction of the UDT experiments.

If the authors are able to address the above questions and requests, then I am more than happy to raise my score.

**Experience Assessment:**

I have published one or two papers in this area.

**Review Assessment: Checking Correctness Of Derivations And Theory:**

N/A

**Review Assessment: Checking Correctness Of Experiments:**

I carefully checked the experiments.

**Review Assessment: Thoroughness In Paper Reading:**

I read the paper thoroughly.

---

> ### Author Response · Authors · 2019-11-10
> **Response to Review #2**
>
> We thank the reviewer for the dedicated review and for recognizing our “impressive empirical results”. The reviewer raised several valid points, which we believe can be easily addressed.
>
> “regularizing with KL-divergence leads to posterior collapse”: This phenomenon was observed in all our datasets (e.g. Cars3D, CelebA), not just in SmallNorb. However, we agree with the reviewer that this does not imply that posterior collapse always happens but only in the settings that we tested.  We revised the text to clarify this. We believe that after the revision, the scope of our finding is related precisely.
>
> “special case of KL-divergence regularization”: We completely agree. Although the difference is subtle, we have extensively shown that its contribution to disentanglement is significant. This was clarified in the text.
>
> “How much slower is GLO compared to amortized models?”, “How many iterations do you employ on a given minibatch of data when using GLO?“: GLO requires about twice the number of iterations than amortized models for convergence. For each mini-batch, we perform a single gradient step for the generator parameters and latent codes. This was clarified in the text.
>
> “authors should show us the actual visualizations for the amortized models”: We added the visualizations to the appendix (A.6).
>
> Inductive bias of latent optimization: Following the reviewer’s advice we have dug deeper into the inductive bias of GLO vs. amortized models. We trained our latent optimization (no amortization) model and its semi-amortized variant on Cars3D and measured the accuracy of classifying class labels from content codes after every epoch. The change in the amount of class-dependent information contained in the content codes is presented in the appendix (A.4). It can be observed that a randomly initialized content encoder (for amortization) encodes class-dependent information, which needs to be minimized as the training evolves i.e. initial mutual information is high and is decreased in the process of training. In latent optimization, content and class codes are randomly initialized, there is therefore zero mutual information between them. By the end of training, amortized models often do not completely remove the mutual information between the class and content codes and provide entangled representations, while a model trained with latent optimization preserves a very high degree of disentanglement. We hypothesize that achieving a similar degree of disentanglement by amortization requires a more sophisticated objective and a more careful hyperparameter tuning.
>
> “Unsupervised Domain Translation … I recommend that the authors try at least one other dataset”: We have run our method on two more domain translation tasks - (i) male to female (ii) faces to anime. The results are presented in the appendix (A.9), our method performed well on both tasks. In the second task, adding the preliminary clustering step allowed for diverse face to anime translation. For added clarity, we added pseudo code (appendix A.8) precisely detailing our procedure, more explanations were added to the text and clustering code was added to our repository.
>
> We believe that all of the reviewer’s questions were addressed. The reviewer stated that addressing the questions would form a basis for raising the score.

---

> > ### Comment · AnonReviewer2 · 2019-11-13
> > **Request for training and test losses on objective function**
> >
> > Thanks for the response. Can you, as requested, provide the training and test losses (Eq 6 and its decomposition into reconstruction + regularization terms) for all your models?
> >
> > Regarding the clarification of the UDT preprocessing procedure, can you describe how Algorithm 1 (A.8) matches up to what you did for Edges2Shoes? What counts as a class label versus a style label for Edges2Shoes? When performing k-means clustering, did you do that only on the shoes images, on the collection of both shoes+edges images, or something else?

---

> > > ### Author Response · Authors · 2019-11-15
> > > **Response to Review #2 - Part 2**
> > >
> > > Thank you for your response.
> > >
> > > 1. “provide the training and test losses” - We provide the training and test losses in Tab. 5 (added to appendix A.4).
> > >
> > > 2. “Regarding the clarification of the UDT preprocessing procedure, can you describe how Algorithm 1 (A.8) matches up to what you did for Edges2Shoes” - In the task of unsupervised domain translation on the Edges2Shoes dataset, we first perform per-class clustering in which shoe images are clustered into 100 styles, and edge images are clustered into 100 styles as well (although clustering the edges is done for technical reasons and does not affect the results significantly, good results obtained regardless of clustering edge images). We then define the class as the unique label of domain label x style label (For example: 1-100 class ids for the shoe images and 101-200 class ids for the edges).

---

> > > > ### Comment · AnonReviewer2 · 2019-11-15
> > > > **Response**
> > > >
> > > > Thank you for your clarification. I sincerely appreciate the effort you put into responding to my request and will update my score accordingly. I still have some reservations about some of the claims put forth in the paper, but I believe these reservations are outweighed by the experimental merits of the paper.
> > > >
> > > > Regarding the training and test loss, may I ask why Table 5 only shows Cars3D? I would appreciate if the authors can make a commitment to provide the training and test loss for all of the datasets used in the paper.
> > > >
> > > > I think the numbers will be valuable in shedding light on the behavior of GLO v. amortization. For example, I am surprised that you achieve better training loss without amortization than with, especially since you only take a single gradient step on the latent code. This defies conventional wisdom that amortization accelerates optimization. Perhaps with enough gradient steps, single-gradient-step-GLO will ultimately still achieve better training loss than amortization, just as is the case in Table 5. Having access to Table 5 for all of your datasets would be illuminating and also give practitioners are better sense of what to expect when they try out your method versus others.

---

> > > > > ### Author Response · Authors · 2019-11-15
> > > > > **Response to Review #2 - Part 3**
> > > > >
> > > > > We thank the reviewer for the dedicated and fruitful review.
> > > > >
> > > > > Table 5 only shows losses on Cars3D as we have fully assessed the results of the entire ablation study only on Cars3D (as presented in Table 3). As per your request, an extended ablation evaluation on the other datasets (including individual loss values) will be added to the paper.

---

### Decision · Program_Chairs · 2019-12-19

**Decision:**

Accept (Poster)

**Comment:**

This paper proposes a novel method for class-supervised disentangled representation learning. The method augments an autoencoder with asymmetric noise regularisation and is able to disentangled content (class) and style information from each other. The reviewers agree that the method achieves impressive empirical results and significantly outperforms the baselines. Furthermore, the authors were able to alleviate some of the initial concerns raised by the reviewers during the discussion stage by providing further experimental results and modifying the paper text. By the end of the discussion period some of the reviewers raised their scores and everyone agreed that the paper should be accepted. Hence, I am happy to recommend acceptance.